# Assessment of the Impact of PM$_{2.5}$ Exposure on the Daily Mortality of Circulatory System in Shijiazhuang, China

**Guiqin Fu [1], Xingqin An [2,\*], Huayue Liu [1], Yaqin Tian [1] and Pengpeng Wang [3]**

[1]  Hebei Meteorological Service Center, Hebei Key Laboratory of Meteorology and Ecological Environment, Shijiazhuang 050021, China; fgq84@tom.com (G.F.); liuhy12@lzu.edu.cn (H.L.); tyq.excellent@163.com (Y.T.)

[2]  China Academy of Meteorological Sciences, Beijing 100081, China

[3]  Chengde Meteorological Bureau Meteorological Service Center, Chengde 067000, China; wangpeng6093@163.com

\*  Correspondence: anxq@cma.gov.cn

**Abstract:** Air pollution can increase the morbidity and mortality of cardiovascular and cerebrovascular diseases, but there are few related studies in counties and cities with serious pollution in China. China is at a critical stage of environmental pollution control. Assessing the health impact of PM$_{2.5}$ (particulate matter with a diameter equal or lower than 2.5 micrometers) on the death toll from cardiovascular and cerebrovascular diseases in heavily polluted counties and cities is of great importance to the formulation of air defense policies related to PM$_{2.5}$. Generalized additive models (GAMs) were used to analyze the effects of PM$_{2.5}$ exposure on the death toll of circulatory system diseases in 16 districts, counties and cities in Shijiazhuang from 2014 to 2016 after controlling the long-term trend of the time series, seasonal effects, holiday effects, air temperature, relative humidity and other factors. The average PM$_{2.5}$ concentration was 121.2 ± 96.6 μg/m$^3$; during the corresponding period, the daily mean mortality of circulatory system diseases in Shijiazhuang was 4.6 ± 4.7. With the increase of PM$_{2.5}$ by 10 μg/m$^3$, the risk of total death from circulatory system diseases with a lag of two days (lag02) increased by 3.3‰ (95% confidence interval (CI): 1.0025, 1.0041). The relative risk (RR) of the effect of PM$_{2.5}$ exposure on the death toll of the circulatory system in Shijiazhuang is consistent with the spatial distribution of the PM$_{2.5}$ concentration and the mortality of circulatory system diseases: the RR of the eastern plain with heavy pollution and a relatively dense population is high, while the RR of the western mountainous area with relatively light pollution and a relatively sparse population is low. For every 10-μg/m$^3$ increase of PM$_{2.5}$, the risk of the increasing death toll from circulatory system diseases in Luancheng of the eastern plain is the highest at 11.9‰ (95% CI: 1.0071, 1.0168), while the RR of Pingshan of the western mountainous area is the lowest at 2.1‰ (95% CI: 0.9981, 1.0062). Conclusions: Based on the epidemiological analysis and GAMs model, after controlling for other confounding factors, PM$_{2.5}$ exposure increased the death risk of the circulatory system in Shijiazhuang, and the risk is higher in heavily polluted plain areas. It provides a scientific basis for formulating scientific air pollution prevention and control policies and provides a reference for improving the prevention awareness of sensitive groups.

**Keywords:** circulatory system mortality; PM$_{2.5}$ exposure; health risk assessment; Shijiazhuang

## 1. Introduction

Since 2013, there has been a lot of persistent large-scale haze pollution weather in China, and the impact of PM$_{2.5}$ (particulate matter with a diameter equal or lower than 2.5 micrometers) fine particles on human health has become an important social issue in China [1–5]. According to the Global

Burden of Disease (GBD) 2010 assessment [6], 3.2 million premature deaths per year are attributed to environmental $PM_{2.5}$ exposure. In China, fine particulate pollution ranks fourth among the main risk factors, and 1.2 million premature deaths each year are related to it [7,8]. $PM_{2.5}$ pollution is one of the main risk factors leading to premature death [9].

Epidemiological studies show that air pollution has significant negative effects on human health; especially fine particulate matter ($PM_{2.5}$) may cause obvious damage to human health. Previous studies focused on Beijing [10,11], Tianjin [12], Shanghai [13] and Guangdong [14] and other major cities in China. Previous studies about the impact of $PM_{2.5}$ on residents' health show that human $PM_{2.5}$ exposure can lead to ischemic heart disease (IHD), ischemic stroke, cardiovascular disease and health and mortality effects [15]. Research shows that cardiovascular disease mortality increases 0.04% with a 10-ug/$m^3$ increasing $PM_{10}$ concentration [16]. Chen et al. studied the relationship between $PM_{10}$ concentration and daily mortality in 16 cities in China from 1996 to 2008 and found a significant correlation: Cardiovascular disease mortality increases 0.44% with a 10-ug/$m^3$ increasing of a two-day lag $PM_{10}$ concentration [17]. In recent years, air pollution in Eastern China, especially in the Beijing-Tianjin-Hebei region, was also quite serious, and previous researches primarily focused on $PM_{10}$; there are few studies on human health impacts related to $PM_{2.5}$ exposure in these areas.

Shijiazhuang City is located in the North China Plain, adjacent to the Bohai Sea in the east, Taihang Mountain in the west and about 270 km away from the capital of Beijing in the north. Its special topography and meteorological conditions are not conducive to the diffusion of air pollutants, making it one of the most polluted regions in China. Assessing the impact of $PM_{2.5}$ in severely polluted areas on the death toll of cardiovascular and cerebrovascular diseases is critical to formulating relevant prevention and control policies.

The Ministry of Ecological Environment started the real-time monitoring of $PM_{2.5}$ concentration at the county level since 2014. This paper collected and collated the monitoring data of $PM_{2.5}$ concentration from 2014 to 2016 published on the website of the Ministry of Environmental Protection of China and the data of disease deaths in the circulatory system diseases of all districts and cities in Shijiazhuang at the same time. Based on the epidemiological analysis, the impact of exposure of the $PM_{2.5}$ population on the death toll of circulatory system diseases was evaluated by using a generalized additive model and nonparametric binary response model, providing a scientific basis for formulating scientific prevention and control policies of air pollution, so as to provide a reference for raising the awareness on prevention of the sensitive population.

## 2. Shijiazhuang Air Quality Data and Methods

### 2.1. Study Area and Air Quality Data

Shijiazhuang (37°27'—38°47' N, 113°30'—115°20' E) has a total permanent population of 10.6162 million (2015) [18] and is the capital city of Hebei Province. From 1 January 2014 to 31 December 2016, the daily death toll of circulatory system diseases in Shijiazhuang came from the Shijiazhuang Disease Prevention and Control Center, including deaths caused by coronary heart disease, ischemic heart disease, ischemic stroke, cerebral hemorrhage, cerebral infarction and other diseases. The average annual resident population data comes from the official website of the Shijiazhuang Statistics Bureau [18].

The monitoring data of the $PM_{2.5}$ daily average concentration on from 1 January 2014 to 31 December 2016 are from the website of the Ministry of Environmental Protection of China. The daily average temperature, relative humidity and other meteorological data of the same period are provided by the Hebei Meteorological Information Center.

### 2.2. Methods

The daily death toll of circulatory system diseases is counted according to a time series. The impact of $PM_{2.5}$ exposure on the death toll of circulatory system diseases is evaluated by using the Poisson

distribution generalized additive model (GAM). All data analyses were conducted using R 3.4.3 statistical software with the "mgcv" package and EmpowerStats to fit the regression models [19].

In this paper, a basic model (Model I) was first established to explore the impact of a single factor ($PM_{2.5}$ exposure) on the daily death toll of circulatory system diseases. Then, Model II added a control of the long-term time trend, seasonal change and holiday effect using the regression spline function based on Model I. Additionally, in order to eliminate the mixed effects of meteorological elements, the daily average temperature (T) and relative humidity (RH) are introduced into Model III based on Model II. The partial auto correlation function (PACF) is used to select the degree of freedom of the time trend until the absolute value of the PACF sum reaches the minimum, and the degree of freedom is selected based on minimization. Akaike's information standard (AIC) and the residual analysis are used to evaluate the selection of the degrees of freedom (df) [20]. The final model is described as:

$$\text{Log } E(Yt) = \alpha + \beta PM_{2.5} + s(time, df = 8) + s(T, df = 4) + s(RH, df = 4) + \text{Season} + \text{Holiday} \quad (1)$$

In the formula, $E(Yt)$ is the expected number of deaths from circulatory system diseases on day t, $\alpha$ is the intercept, $\beta$ is the regression coefficient, s ( ) is a nonlinear spline function and df is the degree of freedom.

All results were expressed by the relative risk (RR) and 95% confidence interval (95% CI) of the death toll from circulatory system diseases. In addition, the impact of $PM_{2.5}$ on the different genders of male and female and the $PM_{2.5}$ sliding average lag effect were evaluated, with $p < 0.05$ as statistically significant.

On the basis of the above research, a nonparametric bivariate response model was established to fit the three-dimensional diagram of the synergistic effect of $PM_{2.5}$ and the average air temperature on the death toll of circulatory system diseases. The spatial distribution characteristics of the synergistic effect of $PM_{2.5}$ and average temperature on the death toll of circulatory system diseases are described to observe the effects of the synergistic effect.

*2.3. Quality Control*

The data of the daily death toll of circulatory system diseases in Shijiazhuang comes from the Shijiazhuang Disease Prevention and Control Center. The air pollution monitoring data and meteorological data are from the national certified atmospheric automatic monitoring system and meteorological observation system, and there are no missing measurement records. All of the data can be used in the analysis directly.

## 3. Results

*3.1. Statistical Characteristics of the Number of Deaths from Circulatory Diseases, $PM_{2.5}$ Monitoring Concentrations and Meteorological Factors in Shijiazhuang City*

Table 1 shows the number of deaths from circulatory diseases and the statistical characteristics of the $PM_{2.5}$ concentration and meteorological factors in Shijiazhuang. From 2014 to 2016, the total number of circulatory deaths in Shijiazhuang City was 80,433—among which, 55.4% were males and 44.5% were females. In the past three years, the mean daily mortality of the circulatory system in Shijiazhuang was 4.6, with the highest number being 6.9 in Xinji and the lowest 1.7 in Gaoyi. The air pollution in Shijiazhuang was serious from 2014 to 2016. The average $PM_{2.5}$ concentrations increased from Jingxing's 105.9 μg/m$^3$ to Xinle's 132.1 μg/m$^3$, and the regional average level was 121.2 ± 96.6 μg/m$^3$, which was in the state of moderate pollution all year round. During the corresponding period, the daily average temperature in Shijiazhuang was 14.3 °C, while the average temperature of each county changed little, which ranged from 13.3 °C to 15.1 °C.

**Table 1.** Statistical characteristics of the circulatory system deaths, PM$_{2.5}$ concentration (particulate matter with a diameter equal or lower than 2.5 micrometers) and meteorological factors from 2014 to 2016 (mean ± standard deviation). RH: relative humidity.

| Station Name | Total/People | Male/People | Female/People | PM$_{2.5}$/(μg/m$^3$) | T/°C | RH/% |
|---|---|---|---|---|---|---|
| Shijiazhuang | 4.6 ± 4.7 | 2.5 ± 2.8 | 2.0 ± 2.3 | 121.2 ± 96.6 | 14.3± 10.7 | 59.0 ± 19.5 |
| Jingxing | 2.6 ± 1.8 | 1.4 ± 1.3 | 1.2 ± 1.2 | 105.9 ± 89.9 | 13.9 ± 10.5 | 53.6 ± 20.5 |
| Pingshan | 3.6 ± 3.1 | 2.1 ± 2.0 | 1.5 ± 1.6 | 120.7 ± 99.0 | 14.1 ± 10.6 | 56.6 ± 19.2 |
| Lingshou | 3.2 ± 2.2 | 1.8 ± 1.5 | 1.4 ± 1.4 | 127.4 ± 110.5 | 14.4 ± 10.6 | 54.3 ± 20.0 |
| Zanhuang | 2.6 ± 2.1 | 1.4 ± 1.4 | 1.1 ± 1.2 | 122.9 ± 99.9 | 14.5 ± 10.6 | 56.8 ± 20.7 |
| Yuanshi | 3.4 ± 2.4 | 1.9 ± 1.6 | 1.5 ± 1.5 | 119.0 ± 99.2 | 13.8 ± 10.8 | 63.0 ± 18.7 |
| Gaoyi | 1.7 ± 2.1 | 0.9 ± 1.2 | 0.8 ± 1.2 | 125.5 ± 100.1 | 13.3 ± 10.7 | 67.3 ± 17.7 |
| Luancheng | 3.0 ± 1.9 | 1.6 ± 1.3 | 1.3 ± 1.2 | 121.4 ± 93.4 | 14.6 ± 10.5 | 57.8 ± 19.0 |
| Zhaoxian | 5.2 ± 3.6 | 3.2 ± 2.4 | 2.0 ± 1.9 | 121.2 ± 94.4 | 13.7 ± 10.6 | 62.7 ± 17.8 |
| Zhengding | 4.2 ± 2.9 | 2.2 ± 1.9 | 2.0 ± 1.7 | 126.0 ± 100.8 | 15.1 ± 10.6 | 55.0 ± 19.6 |
| Xinle | 4.6 ± 4.3 | 2.6 ± 2.6 | 2.0 ± 2.2 | 132.1 ± 97.5 | 14.5 ± 10.7 | 57.3 ± 19.7 |
| Gaocheng | 3.0 ± 1.9 | 1.6 ± 1.3 | 1.3 ± 1.2 | 115.7 ± 89.3 | 14.2 ± 10.8 | 60.7 ± 18.5 |
| Jinzhou | 6.1 ± 7.6 | 3.4 ± 4.5 | 2.7 ± 3.5 | 120.8 ± 92.7 | 14.8 ± 10.6 | 56.3 ± 18.3 |
| Shenze | 3.2 ± 2.2 | 1.7 ± 1.5 | 1.4 ± 1.3 | 121.9 ± 89.7 | 13.9 ± 10.7 | 60.2 ± 18.8 |
| Wuji | 5.8 ± 4.6 | 3.3 ± 2.8 | 2.5 ± 2.4 | 124.7 ± 93.6 | 14.3 ± 10.8 | 59.4 ± 18.7 |
| Xingji | 6.9 ± 4.2 | 3.7 ± 2.6 | 3.2 ± 2.3 | 130.5 ± 97.5 | 14.8 ± 10.6 | 56.0 ± 19.6 |

### 3.2. Seasonal Variation of the Number of Deaths in the Circulatory System and PM$_{2.5}$ Concentration

According to the seasonal distribution of the PM$_{2.5}$ concentration in Shijiazhuang (Figure 1a), the PM$_{2.5}$ pollution in the winter is the heaviest in Shijiazhuang, reaching 182.7 μg/m$^3$ on average, followed by autumn, with a relative best of 81.4 μg/m$^3$ in the summer, showing the seasonal distribution characteristics. Figure 1b shows the seasonal distribution of the number of deaths from circulatory diseases in Shijiazhuang during the corresponding periods, which is also the highest number of deaths in the winter, followed by autumn and the lowest number in the summer. The number of deaths from circulatory diseases in Shijiazhuang was consistent with the seasonal distribution of the PM$_{2.5}$ pollution concentration.

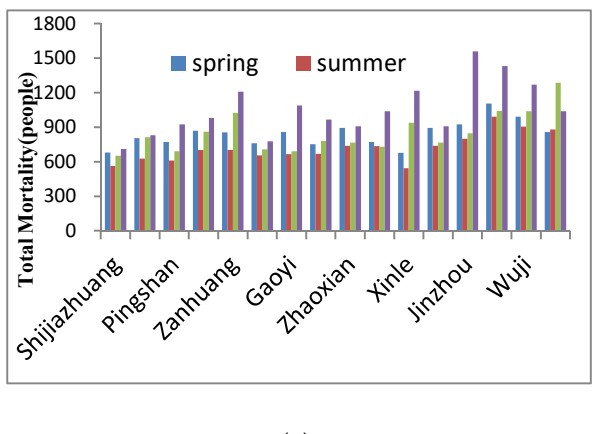
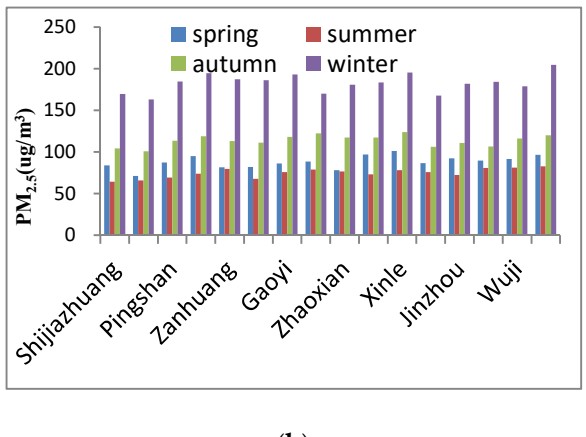

(**a**)　　　　　　　　　　　　　　　　　　　　　　　(**b**)

**Figure 1.** Seasonal distribution of the PM$_{2.5}$ concentration (particulate matter with a diameter equal or lower than 2.5 micrometers) (**a**) and circulatory system deaths (**b**) in Shijiazhuang from 2014 to 2016.

### 3.3. Spatial Distribution of the PM$_{2.5}$ Concentration and Circulatory System Mortality in Shijiazhuang from 2014 to 2016

From 2014 to 2016, the PM$_{2.5}$ concentration in Shijiazhuang decreased year by year, from the annual average of 142.4 μg/m$^3$ in 2014 to 105.0 μg/m$^3$ in 2016. Figure 2a shows the spatial distribution of PM$_{2.5}$ from 2014 to 2016, and it can be seen that the eastern plain of Shijiazhuang with a low-altitude,

flat terrain and a relatively dense population has a high PM$_{2.5}$ concentration, while the western mountainous area with a relatively small population has a low PM$_{2.5}$. The highest concentration of PM$_{2.5}$ is 132.1 μg/m$^3$ in Xinle in the northeast plain, while the lowest concentration is 105.6 μg/m$^3$ in Shijiazhuang and 105.9 μg/m$^3$ in Jingxing in the western mountains. From the distribution of deaths per thousand in the circulatory system in Shijiazhuang from 2014 to 2016 (Figure 2b), it can be seen that the number of circulatory system deaths is also high in the eastern plain and relatively low in the western mountainous area—among which, the highest proportion is 4.57‰ in Shenze, followed by 4.20‰ in Wuji. The lowest death toll was 2.61 per 1000 in Shijiazhuang, followed by 2.90 per 1000 in the western mountainous area of Yuanshi.

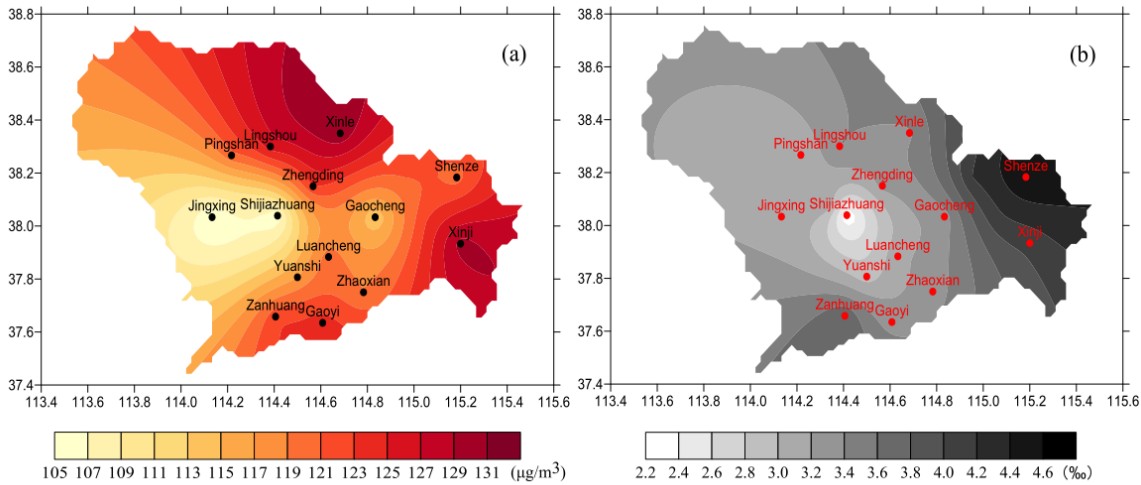

**Figure 2.** Distribution of the average PM$_{2.5}$ concentration (**a**) and circulatory deaths (**b**) in Shijiazhuang counties and cities from 2014 to 2016.

### 3.4. Exposure-Response Relationship between the PM$_{2.5}$ Concentration and Total Circulating Deaths in Shijiazhuang

Worsening circulatory conditions and deaths are associated with long-term exposure to PM$_{2.5}$. Table 2 shows the relative risk (RR) and 95% (CI) confidence interval of PM$_{2.5}$ pollution exposure on the total number of circulatory deaths and the number of males and females of the different genders in Shijiazhuang, as well as the comparison of different regression models. Model I is a single-factor model, model II controls the long-term trend of time changes, as well as seasonal changes and holiday effects and model III considers the temperature and relative humidity variables on the basis of model II. In the three models, whether it is a single-factor model or a model adjusting different covariates, there is no obvious influence on the RR estimation, and the evaluation trends of the three models are consistent. This indicates that there is a stable and consistent relationship between the PM$_{2.5}$ pollution and the number of deaths from circulatory diseases, and PM$_{2.5}$ exposure has the risk of increasing the number of deaths from circulatory diseases. In the fully adjusted model III, for every 10-μg/m$^3$ increase in the PM$_{2.5}$ concentration, the relative risk of the total death toll from circulatory system diseases is 1.003 (95% CI: 1.003, 1.004), which means that the risk of an increased death toll from circulatory system diseases is 3‰. For males, the relative risk is 1.004 (95% CI: 1.003, 1.005) and, for females, is 1.003 (95% CI: 1.002, 1.004). The increased risk for males is slightly higher than that for females, and both passed the significance test of $p < 0.05$.

**Table 2.** The relationship between the PM$_{2.5}$ concentration and the total number of deaths from circulatory diseases and gender differences.

| Projects | Single Factor | Adjust I | Adjust II |
|---|---|---|---|
| Total population | 1.006 (1.005, 1.007) * | 1.002 (1.001, 1.003) * | 1.003 (1.003, 1.004) * |
| Male | 1.006 (1.005, 1.007) * | 1.002 (1.001, 1.003) * | 1.004 (1.003, 1.005) * |
| Female | 1.006 (1.005, 1.007) * | 1.001 (1.000, 1.003) * | 1.003 (1.002, 1.004) * |

Note: * represents $p < 0.05$.

*3.5. Lagging Effect of PM$_{2.5}$ Exposure on the Death Number of Circulatory System Diseases*

After controlling the long-term trends of time change, seasonal change, holiday effects and the influence of temperature and relative humidity (Model III), the influence of different sliding average lag days of the PM$_{2.5}$ concentration on the number of circulatory system deaths in Shijiazhuang from 2014 to 2016 was analyzed (Table 3). As can be seen from Table 3, the PM$_{2.5}$ pollution has a two-day lag effect on the total number of deaths in the circulatory system. At lag02, the relative risk of the total number of deaths in circulatory system diseases reaches the maximum for every 10-μg/m$^3$ increase in the PM$_{2.5}$ concentration, which is 1.0033 (95% CI: 1.0025, 1.0041). Females also have the maximum RR evaluation at a lag02 of 1.0029 (95% CI: 1.0016, 1.0042). For males, the lag0 and lag02 effect values are the same, indicating that PM$_{2.5}$ has a hysteresis effect on the number of circulatory deaths.

**Table 3.** Relative risk (RR) and 95% confidence interval (CI) of the PM$_{2.5}$ exposure and mortality of circulatory diseases in different lagging days.

| Lag Days | Total Population | Male | Female |
|---|---|---|---|
| lag0 | 1.0031 (1.0021, 1.0041) * | 1.0036 (1.0022, 1.0050) * | 1.0025 (1.0010, 1.0040) * |
| lag01 | 1.0025 (1.0014, 1.0036) * | 1.0028 (1.0013, 1.0042) * | 1.0022 (1.0005, 1.0038) * |
| lag02 | 1.0033 (1.0025, 1.0041) * | 1.0036 (1.0025, 1.0048) * | 1.0029 (1.0016, 1.0042) * |
| lag03 | 1.0004 (1.0003, 1.0005) * | 1.0004 (1.0003, 1.0006) * | 1.0004 (1.0002, 1.0006) * |
| lag04 | 1.0004 (1.0003, 1.0005) * | 1.0004 (1.0003, 1.0006) * | 1.0003 (1.0002, 1.0005) * |

Note: * represents $p < 0.05$.

*3.6. Spatial Distribution of the Relative Risk (RR) of Exposure to PM$_{2.5}$ on the Mortality of Circulatory System Diseases*

On the basis of the above analysis, the influence of PM$_{2.5}$ concentration exposure on the number of deaths in the circulatory system on lag02 days was evaluated. Figure 3 shows the spatial distribution of the relative risk (RR) of the number of deaths in the circulatory system related to PM$_{2.5}$ exposure. It can be seen from Figure 3 that the RR was also valued higher in the eastern plain of Shijiazhuang, with a relatively high PM$_{2.5}$. For instance, for every 10-μg/m$^3$ increase in the PM$_{2.5}$ concentration, Luancheng has the highest risk of increasing the total death toll from circulatory system diseases, with 11.9‰ (95% CI: 1.0071, 1.0168), followed by Jinzhou with 9.0‰ (95% CI: 1.0071, 1.0168) and Gaocheng with 8.7‰ (95% CI: 1.0036, 1.0139), while the RR had a relatively low-effect value in the western mountainous area, with a relatively low PM$_{2.5}$. Pingshan in the western mountainous area had the lowest RR value of 2.1‰ (95% CI: 1.0139). For males, the spatial distribution was basically consistent with the total number of people, while, for females, Jingxing was also the relatively approximate center, except for the large RR estimation in the eastern plain, with an RR estimation of 1.0115 (95% CI: 1.0025, 1.0207). The effects of PM$_{2.5}$ exposure on the genders of male and female in different regions were different to some extent.

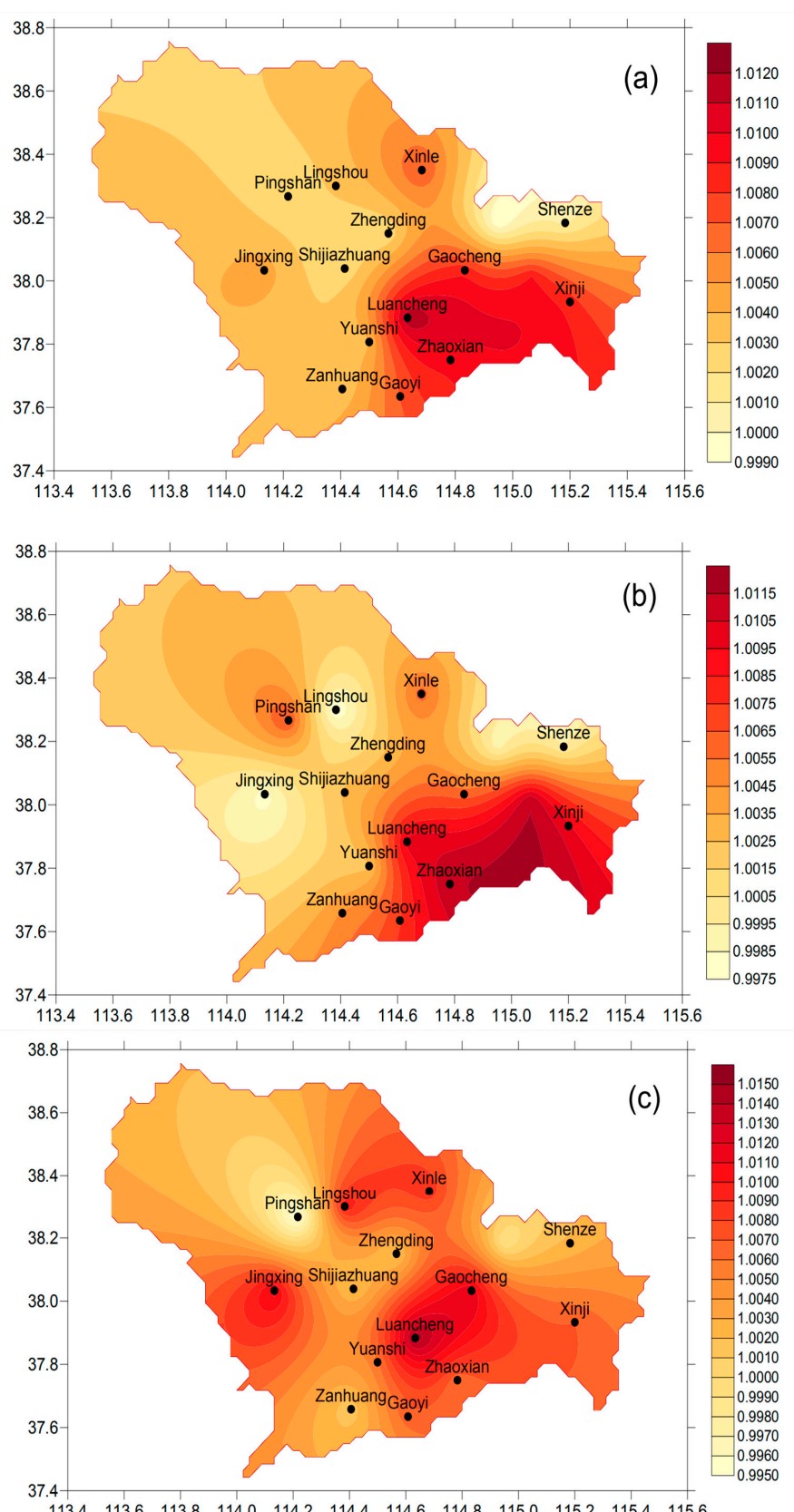

**Figure 3.** Relative risk (RR) of the PM$_{2.5}$ concentration on the number of deaths from circulatory diseases in Shijiazhuang City from 2014 to 2016. (**a**) Total population, (**b**) males and (**c**) females.

### 3.7. PM₂.₅ Synergistic Effect of Exposure and Air Temperature on Circulatory Death

The influence of the average daily temperature on the number of deaths in circulatory system diseases shows a "U" distribution. The temperature is relatively appropriate within a certain range, but too high or too low a temperature will lead to an increase in the number of cases or deaths [3,21]. In Shijiazhuang, heavy PM₂.₅ pollution mainly occurs in the autumn and winter. Therefore, we analyzed whether there was a synergistic effect between the exposure of PM₂.₅ concentration and the daily average temperature on the number of circulatory deaths in Shijiazhuang from 2014 to 2016, and the analysis results are shown in Figure 4.

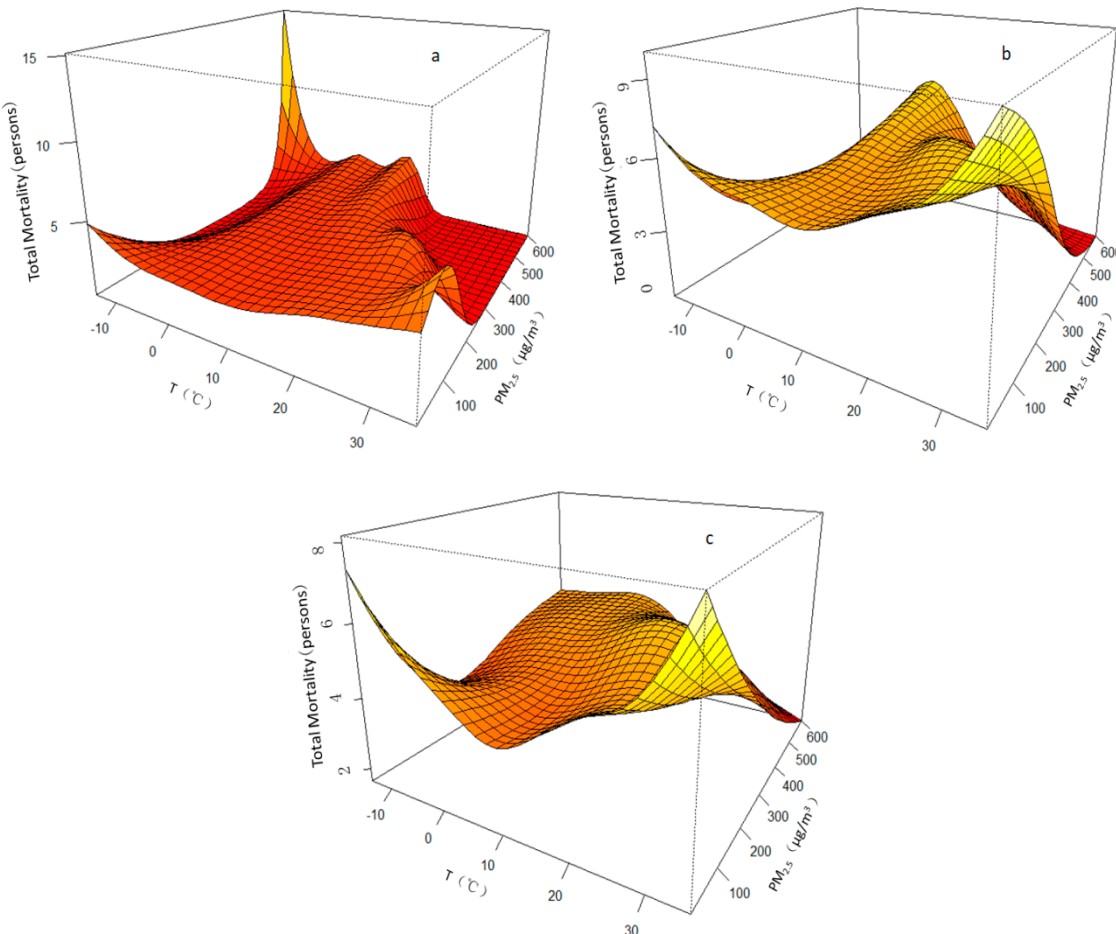

**Figure 4.** Smooth curved surface diagram of the synergistic effect of the daily average temperature and PM₂.₅ on the death toll of circulatory system diseases in Shijiazhuang City from 2014 to 2016. (**a**) Total population, (**b**) males and (**c**) females.

As can be seen from Figure 4, no matter the total number of people or the gender, there is a relatively high value near the high concentration of PM₂.₅ pollution and the temperature of 0 °C. The relatively high number of deaths from circulatory diseases does not occur in the case of the high PM₂.₅ concentration and lower or higher temperatures. This indicates that the synergistic effect of air temperature and PM₂.₅ pollution on circulatory deaths is not clear.

## 4. Discussion

In this study, we used PM₂.₅ concentration data from all counties and cities of Shijiazhuang, China's most polluted city, to explore the relative risk (RR) of PM₂.₅ exposure on the number of deaths due to circulatory diseases in 16 districts and counties of Shijiazhuang from 2014 to 2016 and

whether there is a synergistic effect of $PM_{2.5}$ exposure and temperature on the number of deaths due to circulatory diseases. The results showed that the exposure to $PM_{2.5}$ had an increased risk to the death toll of circulatory system diseases in Shijiazhuang. After controlling the long-term change trends of time, seasonal change, holiday effect and the mixed influence of temperature and relative humidity, it was found that, for every 10-$\mu g/m^3$ increase in $PM_{2.5}$ concentration, the relative risk of the total death toll from circulatory system diseases reached the maximum, reaching 1.0033 (95% CI: 1.0025, 1.0041). Previous studies have shown that short-term exposure to $PM_{2.5}$ is associated with an increase in hospitalization for cardiovascular (CVD) diseases [3,10,12,22]. Shah AS et al. [23] assessed the correlation between global air pollution and heart failure by a meta-analysis method and concluded that, for every 10-$\mu g/m^3$ increase in $PM_{2.5}$, the number of hospitalized patients with heart failure increased by 2.12% (95% CI: 1.42, 2.82). Guo [24] analyzed the influence of $PM_{2.5}$ on the number of CVD emergency patients in Beijing after adjusting the influence of mixed factors such as the temperature and humidity and showed that the relative risk of the number of emergency patients with CVD was 1.005 (95% CI: 1.001, 1.009) for every increase of $PM_{2.5}$ of 10 $\mu g/m^3$.

In the western region of Shijiazhuang, due to the influence of the Taihang Mountain terrain, the terrain is relatively high, and the population is relatively sparse. Besides, due to the prevention and control of air pollution in recent years, the concentration of $PM_{2.5}$ in this region is relatively low, and the risk of increasing the death toll of circulatory system diseases due to the exposure to $PM_{2.5}$ pollution is relatively low. For every 10-$\mu g/m^3$ increase in $PM_{2.5}$, the increased risks of Pingshan, Lingshou, Zanhuang, Yuanshi and Jingxing are 2.1‰, 3.3‰, 3.3‰, 3.9‰ and 4.5 ‰, respectively. However, the eastern plain area of Shijiazhuang is seriously polluted, which increases the risk of mortality, with a range of 2.2–11.9‰. Some studies have shown that, in North China (Beijing-Tianjin-Hebei), the $PM_{2.5}$ pollution level is high, which is attributed to the high mortality rate of $PM_{2.5}$ per 10 $\mu g/m^3$. Shijiazhuang ranks among the top 338 cities in the country [25], which is consistent with the results of this study.

In the impact analysis of the lag effect, this study found that $PM_{2.5}$ exposure has a lag effect of two days (lag02) on the death toll of circulatory system diseases. A coincident result was also found in the study of the impact of $PM_{10}$ on daily mortality, which was done by Chen et al. [17]. Metzger KB [26] and Tolberta PE [27] analyzed the impact of air pollution on the number of daily cardiovascular emergencies in Atlanta and found that the strongest association was usually found on the day. Tan et al. [28] assessed the impact of different air pollutant concentrations on the number of CVD outpatient visits per day and found that $PM_{2.5}$ had the greatest impact on the same day (lag0) or one day later (lag01). It can be seen that, for different outcome variables of emergency and death, the lag effect of $PM_{2.5}$ is also different, because there is still a time process from the emergency medical treatment to death.

In the research on whether $PM_{2.5}$ and the air temperature have a synergistic effect, the synergistic effect is not significant. The primary aspects that affect the death toll of circulatory system diseases are high concentrations of $PM_{2.5}$ pollution and significant cooling due to the influence of cold air in the winter. However, these two extreme conditions cannot occur simultaneously. Since cold air rushes usually along with high winds that can contribute to the diffusion of pollutants, this will lead to a lower concentration of $PM_{2.5}$.

This study assessed the impact of $PM_{2.5}$ exposure on the number of circulatory deaths in Shijiazhuang. In this paper, the $PM_{2.5}$ monitoring concentration data of 16 districts and counties of Shijiazhuang were used. This data were tested and released publicly for the first time in 2014, and the data series lasted for three years. In this paper, the $PM_{2.5}$ monitoring concentration data of 16 districts and counties in Shijiazhuang for three years were used as a data series, which was relatively rare in previous studies. The research results were representative to some extent and could provide reference for other urban areas. On the other hand, the daily death data obtained from the CDC (Center for Disease Control and Prevention Page: 10) are more representative in the regional distribution than the hospital medical records. The limitation of this paper is that the exposure concentration adopted is the

average of each district and county in Shijiazhuang, which fails to accurately reflect the individual exposure situations and may cause some bias.

## 5. Conclusions

This study evaluated the impact of $PM_{2.5}$ concentration on the number of deaths in circulatory system diseases in 16 counties and cities in Shijiazhuang from 2014 to 2016. The results show that the concentration of $PM_{2.5}$ is high in the eastern plains of Shijiazhuang and low in the western mountainous areas; $PM_{2.5}$ exposure has an increased risk of death tolls of circulatory system diseases in Shijiazhuang, and areas with high $PM_{2.5}$ concentrations have a higher risk of death from circulatory system diseases.

**Author Contributions:** G.F.: analysis and writing—original draft preparation, X.A.: writing—reviewing and editing, H.L.: formal analysis, Y.T.: data processing and P.W.: validation. All authors have read and agreed to the published version of the manuscript.

**Funding:** This work was supported by the National Key Research and Development Program of China (2016YFA0602004), and the National Natural Science Foundation of China (41975173).

**Conflicts of Interest:** The authors declare that they have no competing interests.

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
