# Peer review of "Assessment of the Impact of PM2.5 Exposure on the Daily Mortality of Circulatory System in Shijiazhuang, China"

_atmosphere, doi:10.3390/atmos11091018_

Round 1
Reviewer 1 Report
This paper presents an analysis of the impact that PM2.5 exposition has on mortality of circulatory system in Shijiazhuang, China. The analysis is carried out from 2014 to 2016. The results showed that exposure to PM2.5 increased the risk to the death toll of circulatory system in Shijiazhuang. In the impact analysis they identified a lag effect of two days. The research presented in this paper will help to better understand the impact that PM2.5 concentrations have on health. Overall, the manuscript is well written with a good work on the use of references. The paper, which fits well within the scope of Atmosphere, is recommended to be published after working on the following comments.
Abstract. I would be good to read the main findings of this manuscript in the abstract. At the moment it only states the importance of the study.
The introduction gives a good description of the PM2.5 impact on health. Perhaps the authors could add a paragraph about PM2.5 concentrations in urban areas in China. Also, maybe mention previous studies similar to this one in China or other countries for the reader to have an idea about the results you will be presenting.
Results. I might have missed it but I did not read details about the PM2.5 measurements. If possible mention the type of instruments or the time resolution, are they off-line filter measurements, or on-line continue measurements.
The manuscript shows analysis of data from 2014-2016, however, it would be good to see, perhaps in the supplement, statistical analysis of annual data to see the yearly behaviour. Also, it would be nice to see the time series, perhaps with daily values, of PM2.5 concentrations and number of deaths.
Line 20. Change ‘a hot’ for something like ‘an important’
Line 20 Define PM2.5
Line 25. The risk to premature deaths, is that in the world or specific to China?
Line 28. Maybe define PM2.5 as particulate matter with a diameter equal or lower than 2.5 micrometres.
Line 30. Maybe change ‘scholars’ with researchers.
Line 31. Residents of which cities? Or perhaps ‘residents of urban environments’.
Line 67 Edit ‘of on the death toll’ with ‘on the death toll’
Section 2.3 Quality control. The authors mention where they are getting the data, which is okay. However, they should mention if the data was quality assured before shared with them.
Line 108. Edit ‘among which the most number was 6.9 in Xinji and the least was 1.7 in Gaoyi’ with something like ‘With the highest number being 6.9 in Xinji and the lowest 1.7 in Gaoyi’.
Line 109 Change ‘The average PM2.5’ with ‘The average pm2.5 concentrations’
Line 113 What do the author mean by ‘the maximum temperature difference was 1.8℃’? please explain and/or rephrase.
Line 121. Change ‘obvious’ with ‘a’
Figure 1. Are these values average? It is interesting to see a higher increase from autumn to winter on PM2.5 than on deaths. Maybe looking at other statistics or ratios between autumn/winter would show interesting results.
Line 219. Change ‘obvious’ with ‘clear’.
Line 229. Change ‘some’ with ‘previous’
paragraph 247 -250. Please rephrase.
Paragraph 259-264. If I understand well the authors mention that in autumn and winter there are high temperatures, which is not necessarily true. Also, they mention that due to cold air, that atmospheric circulation is ‘good’ which I would not describe it in that way. Due to the low temperatures, there are stagnant conditions in the atmosphere, allowing pollutant concentrations to build up. Please, rewrite this paragraph.
Reviewer 2 Report
The authors provide a nice statistical analysis of the relationship between mortality and PM2.5 concentrations. The work is excellent and I recommend publication.
The authors might double check their presentation and edit as necessary to be certain that it's completely clear.
Reviewer 3 Report
General Comment
The article includes an interesting subject in the impact of PM2.5 Exposure on the mortality of the circulating systems. The authors presented the relationship between PM2.5 exposure and the daily death toll of circulatory system diseases. However, there is some information missing. The authors did not include the environmental conditions separately in all investigated locations for all seasons. Also, the standard deviation with average values of PM2.5 measured in 2014-2016 should be indicated in Fig. (Fig. 1). The introduction needs deep improvement (background, why this paper/work is important, define the purpose and specific purpose and main aim, and the further details on past studies.)
Specific Comment
Below please find the fragments which need improvement:
Abstract
Authors have to take a look at “the instruction for authors”. There is only the situation of China on a critical stage of environmental pollution control. The authors did not present the main results and conclusions.
- Introduction
1) References should be numbered in order of appearance and indicated by a numeral or numerals in square brackets, e.g., [1] or [2,3], or [4–6].
2) The introduction needs deep improvement (background, why this paper/work is important, define the purpose and specific purpose and main aim, and further details on past studies) and include references regarding this study. The lack of existing research and the important issues associating with this study should be included in this introduction part.
- Data and Methods
1) Line 55: Please specify the exact experiment monitoring/investigation duration. Every day from1/1/2014-12/31/2016?
2) Line 61-64: How did you the data analysis? One day average? Or monthly? And please present the method/program for the data analysis.
3) Line 68-69: Why did you select the Poisson distribution generalized additive model?
When you select the model, authors have to explain the reason the selection of the model in the introduction and/or method part and have to compare with other models.
4) Line 73: Please explain the reason for the selection of model 1 and model 2 and need the detail to explain in the Method part.
5) Line 76, 90-91: Please indicate the device for measuring PM2.5, RH and temperature, company, and model….
6) Line 85-88: Please present the program used in the analysis of p-value and if the authors used the statistical program, their program should be explained.
7) Line 94. Did you check the Quality control? Also did Quality Analysis? In my opinion, on the QA and QC analysis, if authors used the air pollution monitoring data and meteorological data are from the national certified atmospheric automatic monitoring system, QA/QC data of this study should be provided at least in the supplementary document.
- Results
1) How did you get the methodological factors? Please rearrange and accurately explain the number of deaths, PM2.5 concentrations, methodological factors. In my opinion, PM concentrations, a number of people died, and factors…this is mainly a case study but the authors also propose new methodologies/models for linking experimental results with PM2.5 exposure, a number of deaths, and methodological factors as health effects. However, the latter aspect is not well explained. The paper shows some general weaknesses.
2) Line 103-126: I suggest that the results part should be rearranged in 3.1 and 3.2 subsections.
3) Line 128-129: Please indicate the standard deviations with the average PM2.5.
4) What is the difference between Fig. 1 and Fig.2? it seems that they have the same data and there are only the differences in the graph types.
5) Line 171-176: Please express as the graph on the data obtained from model 3 and if you need, compare their model 1, 2, and 3 in the discussion part.
6) In Fig. 3 and Fig. 4, the detailed results needed in the result part. Ex) the relationship between temperature, mortality, and PM2.5 concentrations in all investigated regions. Fig. 3 cannot be clearly displayed (also indicated values.)
- Discussion
1) Results obtained in this study should be compared with those in previous studies and more in-depth discussion should be conducted. The discussion needs deep improvement (include references and comparisons with further details on past studies.)
2) The authors should include their implications should be discussed in the broadest context possible in the discussion part and the specific and important findings in Line 265-266. Need improvement on this study purpose “Assessment of the Impact of PM2.5 Exposure on the Mortality of Circulatory system in…”
- Conclusion
There is no conclusion. The conclusion part is mandatory.
Round 2
Reviewer 3 Report
The manuscript has been hugely improved.
However, I would suggest that the manuscript be edited as follows:
Line 14, 16, 17, 21, 22…: PM2.5->PM2.5 Subscript!!
Line 21: need space
Line 31: Authors wrote the general conclusion. Need a specific conclusion on this study based on the additive model and nonparametric binary response model.
================================================
